# A Review: Microbes and Their Effect on Growth Performance of *Litopenaeus vannamei* (White Leg Shrimps) during Culture in Biofloc Technology System

**DOI:** 10.3390/microorganisms12051013

**Published:** 2024-05-17

**Authors:** Bilal Raza, Zhongming Zheng, Jinyong Zhu, Wen Yang

**Affiliations:** School of Marine Sciences, Ningbo University, Ningbo 315832, China; balirana321@gmail.com (B.R.); zhengzhongming@nbu.edu.cn (Z.Z.); zhujinyong@nbu.edu.cn (J.Z.)

**Keywords:** biofloc technology, microbial effects, gut microbiota, shrimp

## Abstract

In the modern era of Aquaculture, biofloc technology (BFT) systems have attained crucial attention. This technology is used to reduce water renewal with the removal of nitrogen and to provide additional feed. In BFT, microorganisms play a crucial role due to their complex metabolic properties. Pathogens can be controlled through multiple mechanisms using probiotics, which can promote host development and enhance the quality of the culture environment. During culturing in a biofloc technology system, the supplementation of microalgae and its accompanying bacteria plays a beneficial role in reducing nitrogenous compounds. This enhances water quality and creates favorable environmental conditions for specific bacterial groups, while simultaneously reducing the dependency on carbon sources with higher content. The fluctuations in the bacterial communities of the intestine are closely associated with the severity of diseases related to shrimp and are used to evaluate the health status of shrimp. Overall, we will review the microbes associated with shrimp culture in BFT and their effects on shrimp growth. We will also examine the microbial impacts on the growth performance of *L. vannamei* in BFT, as well as the close relationship between probiotics and the intestinal microbes of *L. vannamei*.

## 1. Introduction

In large-scale production where animals endure stress, the deterioration of optimal culture conditions can lead to the emergence of diseases, resulting in significant economic losses [1]. Traditionally, disease control in aquaculture, both therapeutic and prophylactic, relied on the use of antimicrobials [2]. Nevertheless, this practice is widely criticized today due to its impact on the accumulation of environmental residues and the development of resistance, which also affects consumer acceptance of the product [1,2]. To improve shrimp nutrition and increase disease resistance, administering microorganisms is an ecologically friendly and safer practice [3]. Furthermore, it requires good aquaculture practices to ensure efficient food utilization by the animals, thereby increasing productivity. Probiotics, which are living organisms, have beneficial effects on the host. They can alter the microbial community associated with the environment or the host, enhance food consumption or increase its nutritional value, boost responses to diseases, and improve the quality of the surrounding environment [4].

As a complex aquatic microcosm, bioflocs consist of microorganisms with diverse metabolic profiles and ecological niches. Among these, autotrophic microorganisms include photosynthetic bacteria, chemoautotrophic bacteria, and microalgae [5]. By using a variety of organic carbon sources, heterotrophic bacteria can degrade residues, which are then transformed into bacterial biomass [6]. In a biofloc technology system, zooplankton typically comprise protozoa such as Sarcodinas, ciliates, and flagellates as well as metazoan such as nematodes and rotifers. A greater variety of biofloc organisms can aid in enhancing the mineralization of waste flocs, [5,6], the consumption of protein [7], and controlling pathogens [8]. The meticulous development of a floc community in biofloc technology depends on several factors, such as the type of external carbon, environmental situations including oxygen levels, salinity, solids load, and light, the carbon to nitrogen (C/N) ratio, and the species being cultured [9].

The ability of an organism to adapt to a particular habitat or set of living conditions is largely dependent on the flexibility of its phenotype. Aquatic species of gut microbiota have a positive impact on an organism’s overall health, including its ability to reproduce, digest food, and maintain a full immune system [3]. Aquatic microorganisms inevitably influence the composition of the intestinal microbiota. Consequently, shrimp are likely to consume biofloc bacteria, which in turn affects the microbial population in their gut and the biological functions of the shrimp. The intimate relationships between bacteria and the continuous flow of water through the digestive tracts of aquatic animals impact their native microflora in an aquatic environment. Additionally, the gut microbiome of aquatic animals is significantly influenced by their dietary regimen [10]. The gut microbiota has emerged as an essential microbiological research topic because of recent research efforts focused on understanding the relationship between the gut microbiota and the host, particularly the positive effects of the gut microbiota on aquatic animals [3]. Accordingly, several studies have been directed on aquatic animals, and the findings of these studies have confirmed that controlling the composition of microbes of the gut has positive effects on the growth, health, and survival of several aquatic animals, including the Pacific white shrimp [3,11].

Globally, 70% of cultivated shrimp are Pacific white shrimp (*Litopenaeus vannamei*), making it one of the most farmed shrimp species. Due to increased cultivation, *L. vannamei* has become a prominent and widespread species, attracting numerous scientific investigations [12]. Additionally, in a zero-water exchange tank system, researchers examined the impact of various biofloc levels on microbial activity, the growth performance of *L. vannamei*, and water quality. The system observed a slower growth rate when the solid concentration exceeded 800 mg/L [12]. The comparison of the gut microbiota and water microbial community between the biofloc system and the clear water system also showed that the biofloc can alter the composition of the intestinal bacteria in shrimp [11]. In general, we will explore bacteria related to shrimp culture in bioflocs technology and their impact on shrimp development. This article will examine the effects of microbes on the growth performance of white leg shrimp cultured using biofloc technology. It will also address the close relationship between probiotics in rearing water and the gut flora of *L. vannamei*.

## 2. Microbes in BFT during Shrimp Culture

Microorganisms are essential to the existence of all living organisms and biotic networks. Various microbial communities in aquaculture systems can have beneficial impacts such impurity exclusion, serving as food for other species, and recycling organic materials. Conversely, they may also have negative impacts, such as causing diseases under harsh environmental conditions [13]. Therefore, it is essential to identify the biological and environmental conditions that influence the function of microorganisms [12]. Several studies have investigated the configuration and structure of biofloc communities in shrimp culture, categorizing the microorganisms into five overlapping categories: nitrifying bacteria, floc-forming organisms, algae grazers, saprophytes, and pathogenic bacteria (e.g., *Vibrio* spp.) [14]. Microorganisms associated with floc-forming extracellular polymeric substances exhibit adhesive properties, which facilitate the formation of cohesive flocs [15]. For a biofloc technology system to function effectively and to achieve its objectives, species from each category must be present, perform their respective roles, and engage in interactions with other species [9].

Ju et al. [16] analyzed the community of microorganisms in Pacific white shrimp farming and observed that the microbial biomass consists of 24.8% phytoplankton (primarily diatom species such as *Navicula*, *Chaetoceros*, and *Thalassiosira*), 4% bacteria, and a small number of protozoa (97% flagellates, 2.0% rotifers, and 1.0% amoebae). The majority of biofloc organisms are heterotrophic bacteria, but other significant species are also present, including algae, ciliates, fungi, rotifers, and flagellates [17,18].

For Pacific white shrimp cultivation in a biofloc technology system, numerous groups of bacteria were observed, but the most abundant species were *Vibrio* spp., *Photobacterium* spp., *Vibrio rotiferianus*, *Marinobacter goseongensis*, and *Proteus mirabilis* [19]. Cardona et al. [20] evaluated the bacterial community found in biofloc technology for *L. vannamei* cultivation and found that species from three phyla, Cyanobacteria, Bacteroidetes, and Proteobacteria, were the most abundant. Proteobacteria comprise roughly 60% of all phyla present at the onset of biofloc formation. These bacteria are widely distributed in seawater and serve a crucial role in the mineralization of organic matter and the recycling of nutrients [20]. The second-most-dominant phylum was Bacteroidetes. These bacteria are the predominant component of heterotrophic marine bacterioplankton and were discovered as macroscopic colonies that feed on organic particles [17]. Cyanobacteria were the third most prevalent phylum of bacterial species [20].

Cardona et al. [20] compared bacterial communities in biofloc technology to those in a water exchange system and discovered that *Leucothrix* constituted approximately 20.1%, *Rhodobacteraceae* constituted approximately 16.4%, *Stramenopiles* constituted approximately 8%, *Oceanospirillaceae* constituted approximately 5.5%, and *Saprospiraceae* constituted approximately 4.5%. Conversely, *Cryomorpgaceae* (about 24.6%), *Pelagibacteraceae* (about 10.1%), *Stramenopiles* (about 8.4%), *Glaciecola* (about 5.6%), and *Colwelliaceae* (about 4.7%) were the sufficient under the “water exchange” condition. These variations can be attributed to variations in biological and physio-chemical parameters of water, such as total ammonia nitrogen, nitrate concentrations, and chlorophyll a level in biofloc technology. In contrast, these parameters remain more stable in the ‘water exchange’ system, influencing bacterial abundances [21]. Mohammad et al. [18] identified the bacterial communities in BFT for *L. vannamei*. They discovered that around 90% of the bacterial community was made up of *Vibrio* sp. (*Vibrio rotiferianus*). There were additionally recognized species of *Bacillus*, *Photobacterium*, *Lactobacillus*, *Clostridium*, *Shewanella*, *Acinetobacter*, *Pseudoalteromonas*, *Alteromonas*, *Pseudomonas*, and *Marinifilum*.

Wei et al. [22] used biofloc technology to detect bacteria using high-throughput sequencing for the 16S rRNA gene. The findings indicated that *Flavobacteriaceae*, and *Rhodobacteraceae* including *Ruegeria*, *Marivita*, and *Maribacter*, were the two major bacterial taxa. Heterotrophic bacteria can be encouraged in biofloc technology aquaculture by increasing the C/N ratio (to more than 10), removing particles, limiting water exchange, and improving aeration and circulation within cultivation tanks. According to Hari et al. [23], the growth rate of heterotrophic bacteria was ten times higher than that of nitrifying bacteria. Heterotrophic bacteria play a role in digesting excrement, dead organic materials, unconsumed feed, and converting nitrogenous ammonia into harmless microbial masses or aggregates. Although they may contribute minimally to denitrification and nitrification, their role in maintaining water quality is significant.

Both heterotrophic and autotrophic microorganisms can be spread using biofloc technology systems [23]. However, the biofloc system contains a greater number of beneficial heterotrophic bacteria (such as *Bacillus*, *Sphingomonas*, *Acinetobacter*, *Micrococcus*, *Nitrosomonas*, *Rhodopseudomonas*, *Nitrospira*, *Pseudomonas*, *Nitrobacter* and *Cellulomonas*) and yeast species. These organisms can support bioremediation and enhance the cultured organism’s performance, water quality, growth, and overall health [24,25,26]. The “core” of biofloc technology is heterotrophic bacteria, which control algae blooms and express system health. According to the “microbial loop concept,” heterotrophs are positioned at the base of the food chain by using dissolved organic matter, recombining up to 50% of the carbon without the interference of microalgae, accelerating mineralization, and making the nutrients available to microorganisms at higher trophic levels. The food chain of biofloc creatures begins with bacteria, which are then eaten by protozoans, which are subsequently eaten by larger organisms [25]. With very high nutrient input loads and minimal water exchange, the reusing process is especially crucial in biofloc technology [27].

### 2.1. Phytoplankton

In intensive shrimp culture systems, the microbial population, which includes bacteria, algae, zooplankton, and other microorganisms, performs critical roles in the cycling of nutrients by supplying nutritional components like fatty acids that are essential to the persistence and growth of shrimp [28]. Phytoplankton serve as the standard feed for shrimp farming in the pond. The increased nitrogen concentration in the pond increases phytoplankton efficiency. The sources of nutrients for phytoplankton growth are metabolic waste and feed [29]. The biomass of phytoplankton determines the biotic structure of an aquatic ecosystem. Understanding the abundance, composition, and succession of phytoplankton is essential for effectively managing aquatic ecosystems. Phytoplankton are the primary producers in these systems and constitute most of ecological pyramids [28]. Being sensitive to changes in water quality, phytoplankton are excellent indicators of the environmental conditions and aquatic health status of ponds. They respond to various stressors, including high nutrient levels, low dissolved oxygen levels, toxic pollutants, predation, and poor food quality [30].

For this review, phytoplankton will be divided into eukaryote groups such as chlorophytes and diatoms and prokaryote groups such as Cyanobacteria. These organisms play a crucial role in the biofloc technology system by consuming nitrogenous compounds to produce sugars and proteins and by releasing oxygen when light is available [31]. In addition to being consumed by zooplankton, phytoplankton transfer nutrients to higher trophic levels. Furthermore, microalgae serve as a valuable nutrient source for aquatic organisms. For instance, eukaryotes such as *Chlorella* sp. and diatoms are the primary sources of essential amino acids and large amounts of polyunsaturated fatty acids for crustaceans [17,30]. Diatom inoculation in biofloc systems enhances the performance like FCR and growth and improves the fatty acid content in the post-larvae of *L. vannamei* [32].

Marinho et al. [28] described that Bacillariophyta improve shrimp development more effectively than the Cyanophyta. They recommended that farm supervisors should prefer a high ratio of green algae and diatoms in the phytoplankton community during the cultivation of shrimp [28]. Green algae are regarded as valuable algae because they act as the feed for fish and most of the aquatic invertebrates. However, Dinoflagellata and Cyanobacteria are related with eutrophication and poor water quality [30]. Marinho et al. [28] also suggested that the most plentiful group was Bacillariophyta and their species which were dominated as *Navicula*, *Coscinodiscus*, and *Skeletonema*. By the end of the experiment, Cyanobacteria, dominated by species such as *Spirulina* and *Oscillatoria*, were the next most abundant group [28]. In the intensive aquaculture system, Lukwambe et al. [30] determined that the number of Chlorophyta and Heterokontophyta increased while the number of cyanobacteria decreased. In contrast, the number of Rotifera and Cladocera increased, while the number of protozoa decreased. The increase in plankton in a community could stimulate the proliferation of shrimp. Administering probiotics changes the composition of phytoplankton in aquaculture ponds. Probiotics’ dynamic activity has been linked to the rise of beneficial algae like diatoms, the consistent presence of key species such as *Coscinodiscus*, and the control of harmful algae [33].

### 2.2. Zooplankton

Numerous zooplankton species related to crustaceans, nematodes, rotifers, and protozoans appear during shrimp culturing in biofloc technology systems and are involved in the maintenance of water quality, the recycling of nutrients, and cultured species nutrition [2,34]. According to Kumar et al. [26] free filamentous, coccoid, and attached filamentous bacteria, *Vibrio* and *Bacillus* species, and flagellate groups, as well as diatoms, ciliates, oocysts, amoebas, rotifers, and nematodes, were among the microorganisms found in the biofloc system to produce shrimp (Table 1). Consuming biofloc microorganisms can enhance feed conversion ratios (FCR), boost growth rates, and provide a more complete diet for cultivated species. Ciliates can reach densities of 40–170 per mL in biofloc systems used for shrimp culture. These findings illustrate that the biofloc system hosts a variety of microorganisms beyond bacteria [35,36]. Rotifers can assimilate 50–60% of raw protein, while Cladocerans have a comparable protein absorption rate of approximately 50–68%. The respective fat contents for these groups are 3.8–13.1% for Rotifers, 1.8–2.9% for Copepods, and up to 2.7% for Cladocerans [16].

Another group of zooplankton, nematodes, also serve as excellent nutrient sources. The dry matter of nematodes consists of 45–50% protein and 15–20% lipids, with the remaining 40% comprising free nitrogen and various macronutrients. Additionally, many ciliates and metazoans feed on bacteria, algae, and fungi [36]. Like flagellates, ciliates are a significant source of free amino acids. Protozoa are an important component of the biofloc system because they can reside within organic particles and consume bacteria [11]. Hargreaves [35] noted the presence of various zooplanktons in the biofloc system, including gastrotrichs, nematodes, euplotes, copepods, rotifers, and ciliates, which operate as links in the food chain between bacteria, algae, and higher species.

Kim et al. [38] investigated the zooplankton composition in a mixed biofloc system for the 42-day culture of *L. vannamei* and a species of red algae named *Gracilaria birdiae*. Rotifers (43.20%), copepods (33.21%), protozoans (13.52%), and cladocerans (10.16%) were among the biofloc species present in the system [38]. According to the literature, rotifers play a crucial role in meeting the nutritional needs of mature species [39]. They are a suitable meal for extremely immature larvae due to their small size range of 0.06–1.00 mm. As a result, it is essential to ensure a sufficient supply of rotifers is available at the right time and place for the survival of many shrimp larvae [37,39]. In the future, the segregation and targeted manipulation of biofloc organisms could enhance shrimp performance in biofloc technology systems, particularly at the larvae or post-larvae stages. This approach could promote growth and reduce the farming period. Therefore, further research is needed in this area [39].

## 3. White Leg Shrimp Microbial Composition

According to the International Council for the Exploration of the Sea (ICES), the ocean is estimated to contain more than 3.6 million microorganisms, which constitute nearly 90% of the total biomass found there [37]. This statistic highlights the diversity of microbial life in the ocean and the extent of its interactions with marine ecosystems [37]. According to Cardona et al. [20], aquatic microorganisms interact with culturing organisms in both mutualistic and pathogenic ways. The 16S rRNA genes of the sample microorganisms were sequenced; the top 10 bacterial genera were determined to be the top genera based on the number of reads. The dominant phyla were those with more than 10% of the readings in a sample. In the residue samples, bacteria were found in 31 phyla, 66 classes, 87 orders, 204 families, and 619 genera, indicating a higher level of microbial diversity than in the water samples, which were categorized into 23 phyla, 44 classes, 69 orders, 151 families, and 398 genera [20].

Additionally, microbial communities changed depending on the sample period. One factor that influences the microbial population in the biofloc technology system is stocking density [40]. At high stocking densities, excess settled organic matter or suspended flocs can be removed through siphoning techniques, settling chambers, or even altering the floc microbial population and affecting the community of microbial floc [40,41]. The abundance of bacterial communities at the genus level was seen at various shrimp stocking densities [40]. At a density of 800 shrimp/m^2^, *Paracoccus* and *Nocardioides* had the highest abundances in the water column, followed by *Rhodopirellula* and *Blastopirellula*, which had the highest abundances at 0.67% and 0.52%, respectively. According to varying stocking density, the water microbial community is clustered into groups [6,41], indicating that density can significantly affect the microbial population in biofloc technology and perhaps alter its functionality. Such fluctuations may be caused by various nutrient input ratios in the cultivation system, which can physically alter the water environment [42]. The genus *Paracoccus* was the predominant genus discovered in the intestines of *L. vannamei* at a density of 600 shrimp/m^2^, accounting for 47.7% of the total population [40]. The dense stocking of 800 shrimp/m^2^ impacted the gut microbiota by shifting the prevalence of *Paracoccus* to 22.44% and elevating *Nocardioides* to 26.64%, indicating a potential decline in the probiotic effect of biofloc at high stocking densities [40].

The diversity of microorganisms is significantly influenced by nutrient accessibility [10,43]. A high stocking density system was found to have the highest gut and water bacterial diversity [40]. However, the nutritional dynamics of biofloc technology are far more complex, considering factors such as shrimp feces, leftover feed, dead microorganisms, and other carbon sources. The accumulation of minerals and other substances in a biofloc system with a high stocking density may lead to further microbial interactions that increase the diversity of the bacteria [20]. Higher floc density may facilitate the aggregation and digestion of shrimp particles at high stocking densities. However, Wasielesky et al. [12] determined that it is the feed protein content, rather than the concentration of higher flocs, which influences shrimp performance.

In the gut of an organism, microbial association is a serious phenomenon that is responsible for several processes, including nutrition absorption, immunological response, homeostasis, and an organism’s overall health [16]. According to Huang et al. [10], gut microbes participate in a wide range of biochemical processes and can be thought of as a metabolically active organ. Ingestion, pathogen defense, and food supply are all critically dependent on intestinal flora. For instance, giving *Bacillus* spp. to the shrimp through the mouth increases its resistance against the disease *Vibrio alginolyticus* [44,45]. Similar to this, a dietary supplement containing live *Bacillus* cells in a weight ratio of 1–5% can increase the digestibility of *L. vannamei*. Despite these investigations, Huang et al. [10] found that we still give the gut bacterial population of *L. vannamei* insufficient attention.

Trillions of microorganisms reside in the digestive system, where the interactions and composition of gut bacteria can significantly impact energy production [46,47]. The intestinal microbiota of cultured aquatic species is crucial for maintaining health. It prevents the colonization of pathogens, aids in food decomposition, generates antimicrobial compounds, releases nutrients, and supports normal mucosal immunity [47]. All gut bacterial communities were found to include members from the three phyla: Actinobacteria, Tenericutes, and Proteobacteria [46]. The results showed that Proteobacteria and Tenericutes were the most prevalent phyla in the intestines of *Litopenaeus vannamei*. Among the six groupings, the core families *Mycoplasmataceae* and *Rhodobacteraceae* ranged from 14.7% to 41.4% and 8.4% to 40.0%, respectively [46]. Even so, Meiling et al. [48] reported that the incidence rate was only 0.01–0.08%. The *L. vannamei* gut was used to isolate nearly 111 bacterial strains, which were then characterized into 13 taxonomic groups, 7 of which were dominant, including the genera *Vibrio*, *Xanthomonas*, *Aeromonas*, *Photobacterium*, *Agrobacterium*, and *Bacillus* and the family *Enterobacteriaceae* [48]. Further research is necessary to confirm whether Proteobacteria are the dominant bacteria in the intestines of *Litopenaeus vannamei*, but they are the most stable within its gut environment.

According to Tzuc et al. [49], *Pseudoalteromonas* and members of the *Vibrio* genera constituted most of the bacterial community in the digestive tract of *L. vannamei* and were distinguished from other bacteria by their capacity to synthesize a wide range of enzymes, including amylases, proteases, chitinases, and lipases. It was discovered that most of these bacteria could produce most of these enzymes, illuminating the involvement of gut microorganisms of *L. vannamei* in the deprivation of food components [50]. Zoqratt et al. [11] compared the diversity of microbes in the gut of *L. vannamei* and rearing pond water in Vietnam and Malaysia and discovered that the shrimp gut had less microbial diversity, with *Photobacterium* and *Vibrio* being the most prevalent bacteria compared to the water around it. According to Ghanbari et al. [42], 90% of the gut microbiome of fish consist of Proteobacteria, followed by Bacteroidetes and Firmicutes, which closely resembles the gut microbiome of shrimp. However, previous studies have shown that the gut microbiome of shrimp varies depending on several factors, including famine, temperature, nutrition, wild versus domestic shrimp species, and developmental stage.

According to previous research, the gut microbiome of *L. vannamei* primarily consists of members of the Proteobacteria phylum, including the families *Vibrionaceae* and *Pantoea agglomerans*, Firmicutes like *Faecalibacterium prausnitzii*, and the phyla Actinobacteria and Cyanobacteria [51]. After Proteobacteria, the most dominant phyla of bacteria are Actinobacteria, Firmicutes, and Bacteroidetes. However, the relative abundances of these organisms vary in the intestine of *L. vannamei* depending on the environment and the diet [51]. Using Solexa sequencing, Yan et al. [52] reported that the abundance of Bacteroidetes was nearly comparable to that of Tenericutes and Proteobacteria in the intestine of *L. vannamei*. Actinobacteria are typically prominent in *L. vannamei*, which is consistent with earlier studies showing that they are frequently found in the guts of shrimp [20]. Additional research is needed to provide concrete evidence that the presence of Actinobacteria in *L. vannamei* is an adaptation to the freshwater environment [13].

The gut of healthy *L. vannamei* has both pathogenic and beneficial microorganisms. Pathogenic bacteria, which can cause disease, are sometimes harmless members of an animal’s normal intestinal microflora but can become harmful under specific conditions. Several opportunistic pathogens may be found in the gut of *L. vannamei*, such as bacteria from the families *Aeromonas*, *Pseudomonas*, *Flavobacterium*, *Rickettsia*, *Shewanella*, *Vibrio*, *Escherichia*, *Desulfovibrio*, and *Enterobacteriaceae* [53]. The abundance of these bacteria in the intestines of healthy *L. vannamei* is either low or balanced in relation to other microorganisms in shrimp. However, during adverse conditions like stress or starvation, the quantity of beneficial bacteria may decrease, potentially creating an environment conducive to the proliferation of unethical or pathogenic bacteria in the intestine, thereby increasing the risk of disease outbreaks [37]. In summary, *Commamonadaceae* bacterial members dominated the intestine bacterial community throughout the early phases of shrimp growth, possibly due to transmission from *Artemia nauplii*. The community was dominated by *Flavobacteriaceae* and *Vibrionaceae* during the middle growth phases and by *Vibrionaceae* towards the final stages of growth. *Flavobacteriaceae* and *Rhodobacteraceae* were found in all stages of development and are most likely the microbiome’s gut core. During the growth stages of *L. vannamei*, the gut bacterial population underwent dynamic alterations at the OTU level [48].

Possessing a diverse array of gut microbiota species may enhance the body’s ability to combat potentially harmful invaders, as it fosters a greater number of species-to-species antagonistic interactions. Early symbiotic species may turn pathogenic when the abundance of specific bacterial types in the microbiome is reduced [11]. It is frequently suggested that bacterial diversity loss within the gut or the differential abundance of microbial taxa may be responsible for the start of pathogenesis because of the connections between the host immune system and the gut microbiota. However, without additional research on gnotobiotic species and gut augmentation, it is often challenging to distinguish cause from effect. Gut microbiotas are frequently recognized as distinct from the bacterial communities in their raising waters [54]. According to Meiling et al. [48], descriptions of shrimp gut communities were based mostly on microbiological techniques and were culture dependent. They suggested that just one or two phylogenic groups dominate shrimp guts and have lesser diversity. However, analyses employing molecular methods indicate that 10–50% of the population is cultivable [48].

White leg shrimp grown in ponds and those raised indoors have highly different gut bacterial populations based on taxonomic profiles (*p* < 0.05). Proteobacteria was consistently the most prevalent phylum in all the samples that were tested, while indoor samples had much greater numbers than pond samples [14]. More pronounced differences were detected between the two aquaculture environments at the family level, with the *Rhodobacteraceae* family accounting for 84.4% of the bacterial communities in indoor samples and the *Vibrionaceae* family accounting for 44.8% of the samples from ponds [49]. In addition, three more groups, Cyanobacteria, Fusobacteria, and Firmicutes, were found in considerably higher abundance in pond sample concentrations than in indoor samples (Table 2). This contrasted with Actinobacteria, which were more frequently detected in indoor samples of roughly 3.0% and 0.06% [48], respectively. A qualitative evaluation of abundance values revealed that the comprehensive taxonomic composition of wild-caught shrimp resembled that of pond-raised shrimp more closely than shrimp cultivated in indoor facilities (Table 2). However, due to the limited number of samples from wild-caught shrimp, a statistically grounded comparison was not feasible [49].

According to Tzuc et al. [49] variations in shrimp gut microbiomes are a result of differences in biology, such as shrimp strains, variations in environmental factors or farming practices (probiotic, temperature, wild capture, diet), or even biases from various laboratory procedures, such as the sequencing platform and the varied fractional 16S sequence target. The assumption in assessing a pond’s microbiome is based on the important roles that microbial communities have begun to play and the potential use of the pond’s microbiome for health surveillance. A comprehensive array of physical and chemical measurements would allow us to correlate the relationship more accurately between rearing water quality and microbiomes, thereby enhancing our understanding of aquaculture microbiomes. Understanding the dynamics of intestinal bacteria during disease development may make therapeutic interventions that normalize the gut microbiota to support and maintain host health easier [36].

The bacterial community exhibited significant variation across different areas and health status groups. The composition of the shrimp’s intestinal microbiota was significantly associated with the water in which they were cultivated. Specifically, the shift in intestinal microbiota between healthy and diseased states did not occur randomly but rather manifested as a gradient. These deviations in intestinal bacterial communities are closely linked to the severity of shrimp disease. This study provides important recommendations for utilizing the appropriate probiotics to rectify the abnormalities in gut microbiota and prevent possible infections in shrimp ponds [55].

## 4. Association between Water and Gut Microbial Communities

In addition to animals found in the terrestrial environment, environmental microorganisms have a significant impact on the bacterial makeup of aquatic animals since they are constantly in touch with ambient water and sediment [56]. For instance, the gut bacterial composition of shrimp was more closely related to that of the water used for culturing [31], although more recent studies have indicated that the microbiota of the intestine of *L. vannamei* is related to the microbiomes in the rearing water or sediment [56,57].

There is currently little information available about the relationship between the bacterial communities in the two different media, including the water and the intestine, in similar ponds of *L. vannamei* [57]. Some leading genera which have a relative abundance greater than 1% in the intestine of shrimp, such as *Acinetobacter*, *Photobacterium*, *Exiguobacterium*, and *Pseudomonas*, were barely examined in water. Similar to this, six groups were divided into two categories, particularly the intestine, sediment, and water samples, through a cluster analysis at the genus level [56]. This demonstrated that, in contrast to the water and intestine, the main bacterial genera in the sediment and shrimp intestine had remarkably similar profiles [57]. It shows a close connection between the bacterial community in sediment and the intestinal microorganisms of *L. vannamei*. In contrast to other environments, the continuous addition of organic matter in the form of feed and other human involvements always affect the bacterial community composition in shrimp aquaculture ponds [58]. As a result, it was assumed that as shrimp farming progressed, the connection between the microbial community and microflora would change [56]. Despite the absence of any statistically significant differences, the strength and variety of the bacterial species in the samples were organized in the order that maximal density was seen in the sediment, then in the water, and finally in the intestine. Because of the apparent increases in detected species, culturing times had a significant impact on the bacterial species abundance in the shrimp’s intestine in relation to water and sediment [43].

## 5. Probiotics

The bacterial ecology within farmed shrimp is significantly influenced by the composition of bacterial communities in ponds during aquatic cultivation; this factor is crucial for their immunity, nutrition, and disease resistance [41]. According to earlier research, seven genera of beneficial bacteria, including *Lactococcus*, *Acetobacter*, *Bacillus*, *Streptococcus*, *Rhodopseudomonas*, *Bdellovibrio*, and *Bacteroides*, have been identified in the intestine of *L. vannamei*. These microorganisms constitute 2.9% of all bacteria, with *Lactococcus* comprising approximately 1.02%, *Streptococcus* around 0.93%, and *Lactobacillus* making up about 0.48%. *Bacillus* bacteria also accounts for a significant portion (0.38%) of all beneficial bacteria found in *L. vannamei*. Nisin, a component of *Lactobacillus* bacteria, exhibits antimicrobial activity [59]. *Lactobacillus* can also prevent the spread of pathogens by maintaining a very low pH in a moderate environment [60]. According to Ding et al. [50] *streptococcus thermophilus* bacteria can prevent the growth of pathogens in the colon and normalize the immune system’s reaction to intestinal inflammation. By producing a variety of digestive enzymes, bacteria in the genus *Bacillus* can enhance nutrient digestion and absorption. To maintain intestinal health and effectively consume nutrients, these bacteria may be essential to *L. vannamei* [61].

When administered in the right amounts, probiotics are live bacteria that provide health advantages to the host [4]. These bacteria have the capacity to colonize and multiply in the gastrointestinal tract, promoting the efficient regulation of a variety of biological systems in the aquatic host. They can be given by water or as feed essences, either alone or in conjunction with other probiotics [45]. A bacteria must be capable of enduring passage through the intestinal tract to be considered a probiotic. It must then spread throughout the digestive tract of the host, either by adhering to mucous membranes or the gut epithelium. Finally, it must be able to reproduce and produce compounds that are antagonistic or suppressive of unwanted instinctual flora [62]. Because they have been associated with marine organisms and have the potential to create a wide variety of chemicals with disinfection activity, *Pseudoalteromonas* species are fortunate in this regard. As a result, pathogen adherence decreases because they can compete with other microorganisms for nutrition and colonization surfaces. The recent findings demonstrate that various exoenzymes are produced by the characterized strains. Given their numerous advantages for the digestive tract of *L. vannamei*, this suggests that they might serve as probiotics in the future [50].

The survival and growth of white shrimp were undoubtedly hampered by the administration of a bacterial strain mixture (*Bacillus* and *Vibrio* sp.), which also had a protective effect against the pathogens *V. harveyi* and the white spot syndrome virus [63,64]. By enhancing phagocytosis and antimicrobial activity, the immune system is supported, which fortifies the body. Probiotic immunomodulation is seen as a collaborative effort involving the introduced microbe, host, and commensal organisms. Through specialized receptors known as pathogen pattern recognition receptors (PRRs), the host can determine whether an organism is infectious or not [41,62].

Supplementing with probiotics may enhance intestinal competitiveness and strengthen the body’s natural defenses against harmful microorganisms [45]. Supplemental bacteria can also adversely influence and combat infections. When *Artemia*, *P. monodon*, and *L. vannamei* are exposed to pathogenic *Vibrio* strains, *Streptomyces* spp. have been shown to have a defending effect, with an increase in endurance described for all three shrimp species [13,57]. The addition of *Streptomyces* sp. *RL8* alone or in combination with *Bacillus* and *Streptomyces* spp. boosted bacterial diversity in the intestine of *L. vannamei* and increased the richness of intestinal bacteria that produce antimicrobials [65,66]. Several probiotic complexes are currently being marketed to the agricultural sector, although administering overall combinations may not be beneficial to the host [67].

Probiotics must initially be able to survive passage through the intestine. According to Adel et al. [68] communal probiotic mixes used in shrimp aquaculture often contain bacterial species that are not native to the marine environment and thus have poor proliferation potential. Candidate probiotics from shrimp intestines can be identified, reducing ambiguity regarding their ability to survive in the host environment, as demonstrated by *Lactobacillus plantarum MRO3.12* [61]. However, the influence of host genetics is a significant factor that restricts the potential of probiotics to modify the composition of intestinal bacteria without proper authorization [59,61]. The use of probiotics should be strictly regulated, even though they have remarkable potential as an alternative to broad-spectrum antibiotics. For instance, probiotic supplements have been found to contain antibiotic-resistant genes [45], including those frequently used in shrimp production [50]. By employing 3.5% and 4.5% supplements and augmented growth due to higher feed consumption, the addition of the macroalgae *Porphyra haitanensis* was previously linked to increasing endurance after WSSV tests [29]. It can be challenging to differentiate between any developmental or health benefits arising from improving gut microorganisms and simply increased nutrient delivery from the addition of supplementary meals [41].

### 5.1. Probiotics Action Mechanism

Probiotics play a critical role in disease prevention in the modern era of aquacultural practices, particularly in biofloc technology. They introduce specific compounds that act proactively against the pathogenic microorganisms, thereby preventing their proliferation in the host organisms [30]. The production of one or more antibiotics [69], bacteriocins [70], lysozymes, siderophores, proteases, hydrogen peroxide, or changes in pH values, may all contribute to the anti-pathogenic activity. Bacteriocins, which are inhibitory chemicals produced by LAB, stop microorganism development [70]. Few bacteria generate substances in addition to these bacteriocins, and these additional compounds are useful for restricting the action of pathogens in the host body [70]. Probiotics enhance the body’s defense against pathogens by mitigating the negative effects of antibiotics and chemotherapy drugs. They boost shrimp immunity, leading to enhanced resistance and higher survival rate across various developmental stages, such as larvae and post-larvae [62]. Varying probiotic techniques that have an impact on antibodies, immune cells, acid phosphatase, antimicrobial peptides, and lysozyme produce varied stimulatory effects on the immune system of shrimp [71]. The activation of antibodies and macrophages enhances the immunological response. The addition of probiotic bacteria to the diet of bait and herring larvae has enhanced immune system stimulation, consequently improving resistance to infections [69].

### 5.2. Probiotic and Pathogen Association in Biofloc

Both the shrimp’s intestinal track and the water utilized for shrimp farming are believed to include a variety of both beneficial and harmful bacteria [68]. Critical factors include the fluctuation of the microbial community in terms of its abundance, diversity, and composition under these conditions, as well as the combined effects of introducing external probiotic bacteria and the microbes present in biofloc water. Understanding this connection is essential for developing strategies for disease prevention, water quality management, and microbial community supervision in biofloc culture systems [39]. This interaction could occur between probiotics and biofloc water pathogens or between probiotics and the gut pathogens of white leg shrimp [68].

To investigate the effects of a *Bacillus* probiotic mixture on the abundance of the presumed pathogenic *Vibrio*., Hostins et al. [41] conducted an experiment in the presence of a BFT culture system. *Bacillus subtilis-complex* and *Bacillus licheniformis* strains were combined in the commercially available probiotic cocktail (Sanolife PRO-W^®^, INVE Aquaculture, Salt Lake City, UT, USA) used in this investigation, with a bacterial concentration of 5 × 10^10^ CFU g^−1^. Four treatments with three replicates each were used in this research. Biofloc without probiotics (BFT), Biofloc with probiotics (BFT + P), Clear Water without probiotic application (CW), and Clear Water with probiotic application (CW + P) were the four treatments [10]. Overall bacterial abundance was much higher in the gut of *L. vannamei* when treated with probiotics than when treated with clear water [56]. The lowest bacterial abundance was seen in the treatments that did not include probiotic supplements (Table 3). Regarding the density of *Bacillus* bacteria, which make up the probiotic mixture, more cells were found in the probiotic-containing treatments [56]. According to Sapcharoen and Rengpipat [60], when considering the proportion of the total bacterial abundance of *Bacillus* and *Vibrio*, *Bacillus* contributed the most significantly to clear water with probiotic treatment, followed by biofloc with probiotic treatments. The influence of *Bacillus* sp. on overall bacterial abundance was observed in the BFT and CW treatments. In contrast, the percentage of *Vibrio* sp. was considerably lower in the treatments that included the administration of probiotic combinations, such as CW and BFT, and significantly higher in those that did not [63].

*Bacillus* typically does not colonize mucosal membranes for very long. By varying the probiotic dose, the bacterial density can be managed [46]. The probiotic can also be consumed by shrimp through its osmoregulatory mechanisms and during the feeding [59]. The formation of probiotics in the shrimp’s digestive tract is closely connected to both the timing and quantity of consumption. As the duration of shrimp feeding increases, the concentration of probiotics in the water also increases [65]. The probiotic *Bacillus* mixture that was put in the biofloc and clear water systems helped the shrimp’s intestines colonize. As a result, adding more probiotic combinations to the water can facilitate the growth of *Bacillus* in the intestine by increasing its numbers [65,71], whereas *Vibrio* can inhibit this growth [63,72].

Aquatic organisms should have a strong correlation between the intestinal microbiota of their host and the microbiome of their surroundings. Thus, probiotics can be used to enhance microbial diversity and manage pathogenic infections in the shrimp’s gut. Past research has shown that using a probiotic, namely *Bacillus* spp., can help increase resistance against *Vibrio* infections. These outcomes were established by challenge tests in vitro antagonistic activity [71], or probable *Vibrio* counts [72]. These contents are almost identical to the recent study in which similar results for the abundance of *Vibrio* and *Bacillus* bacteria in the intestine of *L. vannamei* elevated in clear water with probiotics and biofloc with probiotics were observed. As previously revealed, the specific bacterial quantification of *Bacillus* and *Vibrio* in the intestine of *L. vannamei* demonstrated an antagonistic relationship. The greater numbers of *Bacillus* (BFT + P and CW + P) were related to a lower abundance of *Vibrio*, and vice versa. One of the principal mechanisms of action for probiotics is competing for attachment sites [56].

The production of compounds with qualities like antagonism, bacteriocins, and organic acids is another essential characteristic of probiotics. These chemicals can alter the microbial metabolism [70] and maintain shrimp health by enhancing the immune system, which provides resistance against diseases. The presence and quantity of probiotic organisms in this study enhanced bacterial abundance in the gut of *L. vannamei* in a clear water (CW) and BFT system. This effect undoubtedly influenced the diversity of bacteria and microbial composition, further amplified by the interaction with feed intake and the surrounding environment [50]. The intestinal bacterial community of *L. vannamei* was modified by the adding of a mixture of commercial probiotics containing *Bacillus* bacteria, which increased the diversity of bacteria and decreased the abundance of *Vibrio* in clear water [71]. In this trial, the probiotic demonstrated a robust ability to colonize the digestive tract, and it identified more than 30% of all the bacteria in the shrimp’s gut. Such information could have favorable implications for shrimp, as previously mentioned, such as improved disease resistance and subsequently higher shrimp production outcomes. Whether or not the probiotic mixture was added, the use of biofloc technology as a culture system contributed to an increase in the number of bacteria in the *L. vannamei* gut [27].

In a high-density intensive aquaculture of *L. vannamei*, Hu et al. [73] measured the impact of the combined use of molasses and probiotics (*Bacillus* spp.) as carbon sources on the structure of the microbial community and confirmed that this mixture supported the establishment and development of an advantageous microbial community structure. These results support the findings of an earlier study that found probiotic therapy increased the number of total bacteria in the biofloc. *B. subtilis* and *B. amyloliquefaciens* are two examples of *Bacillus* species that can co-aggregate and combine on their own [60]. Probiotic bacterial strains may require auto-aggregation to adhere to host gut epithelial cells. Like biofloc, co-aggregation may make it possible for external bacteria to combine, allowing a particular biofilm to form. Additionally, probiotic organisms may be able to create a barrier with this property, which could effectively stop pathogen colonization [49].

In clear water and biofloc technology systems, the use of a mixture of commercial probiotics aids in homeostasis and prevents the emergence of opportunistic pathogenic bacteria [74]. According to Aguillera-Rivera et al. [58] a cooperative impact between the heterotrophic bacterial community and the probiotic community for *L. vannamei* may prevent the development of lesions in the hepatopancreas caused by the pathogenic bacteria *Vibrio* in the water. In the biofloc technology system, where probiotic mixture (BFT + P) was present, the overall abundance of bacteria, as assessed in the current study, was higher. However, the results also show that *Bacillus* can colonize the intestine more effectively when *L. vannamei* is grown in clear water than in a BFT system; this is likely due to the biofloc system’s configuration, which involves a wide variety of microbes [75,76]. In contrast, the percentage of *Vibrio* associated with the total microorganisms was lower in the treatment containing a probiotic mixture, aligning with the role of biofloc in regulating or mitigating potentially harmful bacteria, particularly *Vibrio* [72]. These observations demonstrate that the dynamics and interactions among the various groups of organisms constituting the biofloc system can be influenced by the introduction of exogenous bacteria into an already established bacterial community. These recommendations can also be extended to various management strategies for probiotic or biofloc applications. Consequently, there is potential for enhancing the reputation of probiotic supplements, particularly in traditional clear water culture systems [71].

## 6. Microbial Effect on Shrimp Growth

Demonstrating the importance of the gut microbiome in animal health and nutrition underscores the importance of managing their microbial symbionts [77]. Such managements may lead to innovative solutions that can address global difficulties in shrimp farming, including enhancing disease resistance and optimizing growth on economical diets [3]. Achieving this goal requires a deeper understanding of shrimp microbial communities and the influence of factors such as diet, host characteristics, and environmental conditions on the composition of these communities [3]. The manipulation of nutrients via beneficial microbial communities in the intestine of shrimp may provide solutions to increase pathogen resistance without using antibiotics as well as enhancing the development of alternative protein sources. This is because the gut microbiome plays a major role in animal health [77]. According to the findings of the study, Proteobacteria predominantly comprise the gut bacterial profile of healthy shrimp. Undoubtedly, these latter phyla have also been identified as minor components of the shrimp intestinal microbiota, with their prevalence strongly influenced by local environmental conditions and diet [46].

According to Cornejo-Granados et al. [43] there is currently limited understanding of how aquaculture practices impact the gut bacteria of shrimp. For example, biosafety measures may inadvertently diminish or eliminate the colonization of indoor-grown shrimp by beneficial bacteria, which are common in natural habitats, to guard against pathogen infections [43]. Since early intestinal microbial colonization can influence future shrimp performance and efficiency, the growth of healthy microbes may be adversely affected in shrimp cultivated in aquaculture. There is evidence that dietary components can introduce specific microorganisms into shrimp; indeed, it has been reported that the intestinal bacterial communities of shrimp adapt to commercial diets [68].

The improvement in growth achieved through dietary probiotic supplementation has previously been attributed to biological and physiological alterations in the gut environment, as well as structural changes in the intestinal epithelium [64], such as an upgraded intestinal microvillus structure and a higher absorptive surface area [78]. Microorganisms may influence culturing conditions positively or negatively. Some positive effects include the removal of metabolic toxins, such as nitrogen, ammonium, and other substances, from the water column or sediment deposits. Other positive effects include the removal of organic matter from unused feed, bodily waste, deceased organisms, and other sources, along with the recycling of their nutrients back into the system [79] and the enhancement of digestive enzyme performance [49].

Numerous studies have highlighted developmental advancements and improvements in feed conversion ratios, demonstrating that the primary beneficial effect of microbial communities, for practical purposes, lies in their contribution to the nutrition of cultured organisms by converting inorganic nitrogen to microbial protein that was absorbed by the cultured organisms [80]. The presence of microbes in aquaculture tanks increased the efficacy of protein integration from 20 to 25% to roughly 45% [80,81]. In this regard, Hostins et al. [41] discussed the role of microorganisms in aquatic food chains, emphasizing their importance in the feeding of various cultivated species. AHL molecules (N-acyl homoserine lactone) are produced by the pathogenic bacterial species *Vibrio. harveyi* and can be damaged by certain bacterial enhancement cultures that have been isolated from the intestine of Pacific white shrimp. This interference modifies the pathogenic activity by altering signal transmission [82].

The metabolic functions of specific bacterial groups, namely Actinobacteria, Bacteroidetes, and Firmicutes, significantly influence host physiology. For example, these bacteria play a crucial role in fermenting undigestible carbohydrates, leading to the production of short-chain fatty acids (SCFAs) [83]. Notably, Firmicutes synthesize butyrate, an essential compound that promotes the health of the intestinal lining. The phylum Actinobacteria comprises Gram-positive bacteria that exert a probiotic effect through the production of phenol oxidase. This enzyme facilitates the release of mucin, antioxidants, and antibacterial agents, thereby supporting the host’s intestinal health.

Additionally, *Phaeoalteromonas* produces a range of antimicrobial substances [84]. These bacteria are responsible for producing hydrolytic and bioactive enzymes in marine environments and organisms, facilitated by their antifouling, antibacterial, and anti-biofilm activities. In shrimp aquaculture, the probiotic isolates of these bacteria have been employed to mitigate pathogenic infections by *Vibrio* species, leveraging their recognized antimicrobial and probiotic properties.

*Bacillus* bacteria produce highly durable spores that exhibit a resistance to external physical and chemical stresses. Additionally, they synthesize polypeptides such as gramicidin, bacitracin, tyrothricin, and polymyxin, which are effective against a wide range of Gram-positive and Gram-negative bacteria, including pathogenic *Vibrio* strains [55]. Hu et al. [83] observed a decrease in the number of *Vibrio* bacteria in the intestines of *L. Vannamei*, accompanied by an enhanced immune response. This was evidenced by increased total hemocyte counts, serum protein concentration, total antioxidative capacity, and the activities of superoxide dismutase (SOD), proteases, lysozyme, amylase, and phenol oxidase in both the hepatopancreas and intestines of the shrimp. These changes were induced by a diet supplemented with *Bacillus licheniformis*.

Feeding *Streptomyces* strains alone or in combination with *Bacillus* (*Bac-Strep*) enhances the growth rates of *L. vannamei*. The *Bac-Strep* group demonstrated the most potent immunomodulatory effect, significantly enhancing both the total hemocyte count (THC) and superoxide dismutase (SOD) activities. In contrast, the groups fed solely with *Streptomyces*, *Streptomyces* combined with *Lactobacillus* (*Lac-Strep*), and a mixed group (Mix), stimulated only one of these parameters [65]. The modulatory effects of intestinal microbiota on nutrient utilization in *L. vannamei* have not yet been conclusively demonstrated; therefore, the intestinal microbiota of aquatic animals requires further investigation [81,85].

## 7. Conclusions and Future Perspectives

The utilization of energy-saving technologies has significantly simplified modern life by reducing energy consumption. Additionally, the adoption of biofloc technology in aquaculture has positively impacted human life by minimizing water waste and enhancing the availability of nutrients and food. Utilizing microorganisms in biofloc technology represents an eco-friendly approach that enhances water quality, promotes the growth of cultured animals, and eliminates various types of harmful microorganisms. The study of the microorganisms of the biofloc technology system provides a comprehensive understanding of the diversity, abundance, and ecological roles of microbes. This research enables the precise identification of microbial functions in biofloc technology for aquaculture. Microorganisms such as microalgae, bacteria, and other metazoans contribute to the growth enhancement of shrimp in the biofloc technology system. However, the comprehensive diversity of microbes in both the gut of shrimp and the rearing water in the biofloc technology system is still unknown. Consistent with other studies, our research has revealed that the bacterial composition in the water differs from the intestinal bacterial composition of *Litopenaeus vannamei*. According to our findings, Proteobacteria were the predominant bacteria in the water. In the gut of *L. vannamei*, Proteobacteria were also the most abundant group. However, in contrast to Proteobacteria, Cyanobacteria and Actinobacteria did not adapt to the shrimp gut environment. This suggests that the host gut environment imposes selective pressure that influences the establishment of the microbial community. In general, intestinal microbiota should be considered when evaluating the growth performance of aquatic species, particularly white leg shrimp. The intestine of *L. vannamei* is characterized by a selective environment that influences the intestinal bacterial community. The increased prevalence of opportunistic pathogens is primarily due to the dysbiosis of the gut microbiome or an imbalance in the aquaculture microbiome. Intestinal dysbiosis, which leads to diseases in shrimp, adversely affects its growth and development. Certain bacteria have important probiotic applications. The adoption of biofloc technology has also been well received due to its natural approach of utilizing microorganisms for sustainable shrimp aquaculture coupled with its zero-water exchange strategy, which has the inherent property of enhancing water quality. Intestinal microbial manipulations have not only improved the shrimp growth but also bolstered its immunity and gastrointestinal health. Supplementation with beneficial microbial companions and diet-mediated microbiome modulation are poised to be the most effective strategies for managing aquatic health, both now and in the future. These approaches hold the potential to significantly enhance global food production rates. Further research into the effects of gut microbiome and aquaculture microbiome on shrimp health and metabolism, the impact of various dietary formulations on both shrimp and aquaculture microbiomes, and the improvement of BFT through the supplementation of various growth-enhancing substrates for shrimps, would undoubtedly assist the shrimp aquaculture sector in overcoming its current challenges. There are numerous strategies for managing aquaculture systems to cultivate microorganisms that provide the optimal benefits for farmed species. Biofilm and periphyton-based microbial systems and BFT are currently the most widely used microbial systems in the world. This method, as a novel strategy for regulating pathogenic bacteria in aquaculture, requires further investigation. Studies on the community dynamics of microorganisms are essential for expanding aquaculture knowledge for future researchers, farm administrators, and farmers.

## Figures and Tables

**Table 1 microorganisms-12-01013-t001:** List of plankton communities and their effects on *L. vannamei* during the culturing in biofloc technology.

Plankton	Group	Effect on *L. vannamei*	Reference
Phytoplankton	Bacillariophyta	Rich in nutrients like fatty acids, improved shrimp development. Involved in biofiltration, helped to remove nitrogenous waste.	Nasrullah et al. (2018) [29] Lukwambe et al. (2015) [30] Abreu et al. (2019) [32] Martínez-Córdova et al. (2015) [37]
	Cyanophyta	Involved in eutrophication, in result poor water quality, Showed variable effects of growth.	Nasrullah et al. (2018) [29] Lukwambe et al. (2015) [30] Martínez-Córdova et al. (2015) [37]
	Chlorophyta	Different types of vitamins and proteins present, healthy growth.	Nasrullah et al. (2018) [29] Lukwambe et al. (2015) [30] Martínez-Córdova et al. (2015) [37]
	Dinophyte	Involved in eutrophication, some species have negative effect on growth of shrimp.	Nasrullah et al. (2018) [29] Lukwambe et al. (2015) [30] Martínez-Córdova et al. (2015) [37]
	Euglenophyta	Excellent source of protein content, improved immune responses, also improved disease resistance.	Nasrullah et al. (2018) [29] Lukwambe et al. (2015) [30] Martínez-Córdova et al. (2015) [37]
Zooplankton	Rotifers	Can assimilate 50–60% raw protein, enhanced the immune system, increased the survival rate.	Ju et al. (2008) [16] Martínez-Córdova et al. (2015) [37] Ray et al. (2010) [36] Kim et al. (2014) [38]
	Cladocerans	Protein absorption rate is 50–68%, Containing essential fatty acids, Improve shrimp growth rate.	Ju et al. (2008) [16] Martínez-Córdova et al. (2015) [37] Ray et al. (2010) [36]
	Copecodes	Provide beneficial fatty acids, improved the growth rate.	Ju et al. (2008) [16] Martínez-Córdova et al. (2015) [37] Ray et al. (2010) [36]
	Artemia	High nutritional values, (containing omega-3 help in digestion. They also contain proteins and immunostimulants, which enhance the immune system.	Ju et al. (2008) [16] Martínez-Córdova et al. (2015) [37] Ray et al. (2010) [36]
	Nematodes	Serve as excellent nutrient sources, consist of 40–50% protein and 15–20% lipid content, improved the growth rate.	Ju et al. (2008) [16] Martínez-Córdova et al. (2015) [37] Hargreaves [35] Ray et al. (2010) [36]
	Protozoans	Some protozoans are parasites like *vorticella*, Cause water pollution, increase stress in culturing environment, increase mortality.	Ju et al. (2008) [16] Martínez-Córdova et al. (2015) [37] Ray et al. (2010) [36]

**Table 2 microorganisms-12-01013-t002:** Taxonomically, the relative abundance (%) of core bacterial groups in the digestive tract of white leg shrimp cultured in two distinct production systems and from a wild population.

Taxonomy	Indoor	Ponds	Wild	References
Proteobacteria	89.6 ± 3.7	50.8 ± 5.2	62.0	Tzuc et al. (2014) [49]
	90.3 ± 3.2	49.6 ± 4.2	48.3	Meiling et al. (2014) [48]
	86.0 ± 2.1	48.3 ± 5.7	55.2	Sun et al. (2016) [14]
	92.1 ± 7.2	55.0 ± 3.0	57.2	Mohammad et al. (2022) [18]
*Rhodobacteraceae*	82.4 ± 3.6	5.4 ± 6.1	2.8	Tzuc et al. (2014) [49]
	79.3 ± 1.2	6.2 ± 1.8	3.7	Meiling et al. (2014) [48]
	75.7 ± 6.1	6.4 ± 1.1	4.4	Sun et al. (2016) [14]
	80.8 ± 4.1	5.2 ± 2.1	3.3	Mohammad et al. (2022) [18]
*Vibrionaceae*	0.04 ± 0.0	45.8 ± 4.9	56.5	Tzuc et al. (2014) [49]
	1.21 ± 0.1	22.4 ± 3.7	46.7	Meiling et al. (2014) [48]
	0.08 ± 0.0	36.7 ± 6.8	55.2	Sun et al. (2016) [14]
	2.02 ± 0.2	0.08 ± 0.01	49.3	Mohammad et al. (2022) [18]
Firmicutes	4.4 ± 0.8	1.9 ± 0.6	3.7	Tzuc et al. (2014) [49]
	2.7 ± 1.2	1.4 ± 1.0	2.9	Meiling et al. (2014) [48]
	3.3 ± 1.3	1.8 ± 1.1	3.3	Sun et al. (2016) [14]
	4.7 ± 2.1	2.8 ± 0.9	4.0	Mohammad et al. (2022) [18]
Other Proteobacteria	0.6 ± 0.2	37.0 ± 5.9	17.7	Tzuc et al. (2014) [49]
	2.2 ± 0.6	13.6 ± 3.2	12.44	Meiling et al. (2014) [48]
	1.9 ± 1.1	25.6 ± 5.1	14.12	Sun et al. (2016) [14]
	2.0 ± 1.3	22.9 ± 2.6	13.33	Mohammad et al. (2022) [18]
Bacteroidetes	0.01 ± 0.0	8.9 ± 3.4	3.5	Tzuc et al. (2014) [49]
	0.24 ± 0.1	6.7 ± 2.1	4.4	Meiling et al. (2014) [48]
	1.02 ± 0.1	7.0 ± 3.0	3.7	Sun et al. (2016) [14]
	0.28 ± 0.2	8.1 ± 0.8	4.1	Mohammad et al. (2022) [18]
Fusobacteria	2.5 ± 2.2	1.5 ± 0.4	1.2	Tzuc et al. (2014) [49]
	2.8 ± 1.8	2.4 ± 0.9	2.2	Meiling et al. (2014) [48]
	3.3 ± 1.7	3.2 ± 1.2	2.6	Sun et al. (2016) [14]
	3.0 ± 1.2	1.3 ± 0.9	2.9	Mohammad et al. (2022) [18]
Cyanobacteria	2.8 ± 1.2	0.6 ± 0.1	0.1	Tzuc et al. (2014) [49]
	1.4 ± 0.9	1.8 ± 0.4	0.5	Meiling et al. (2014) [48]
	3.6 ± 2.2	1.1 ± 1.3	1.8	Sun et al. (2016) [14]
	2.3 ± 1.8	1.1 ± 0.8	0.8	Mohammad et al. (2022) [18]
Candidatus Saccharibacteria	0.01 ± 0.00	1.7 ± 0.6	7.3	Tzuc et al. (2014) [49]
	0.06 ± 0.01	1.2 ± 0.2	8.4	Meiling et al. (2014) [48]
	0.20 ± 0.02	2.0 ± 0.8	8.6	Sun et al. (2016) [14]
	0.08 ± 0.01	2.2 ± 1.3	7.7	Mohammad et al. (2022) [18]
Actinobacteria	3.5 ± 1.0	0.08 ± 0.1	0.3	Tzuc et al. (2014) [49]
	4.1 ± 1.4	1.02 ± 0.5	0.8	Meiling et al. (2014) [48]
	5.7 ± 2.7	1.80 ± 0.4	1.0	Sun et al. (2016) [14]
	3.2 ± 1.0	0.20 ± 0.1	0.5	Mohammad et al. (2022) [18]

The data shown are the mean and standard error of the mean, in that order. The indoor and outdoor samples showed a statistically significant difference (*p* < 0.05).

**Table 3 microorganisms-12-01013-t003:** *Bacillus* and *Vibrio* species’ proportions to the total number of bacteria found in clear water and using biofloc technology with and without probiotics.

	Biofloc with Probiotics	Biofloc with No Probiotics	Clear Water with Probiotics	Clear Water with No Probiotics	References
*Bacillus* spp.	17.26 ± 4.1	3.58 ± 3.0	32.85 ± 2.4	1.90 ± 1.0	Sapcharoen and Rengpipat (2013) [60]
	19.33 ± 3.1	4.01 ± 2.2	24.43 ± 3.0	2.34 ± 0.9	García et al., 2017 [65]
	24.22 ± 2.1	5.21 ± 1.9	27.60 ± 3.0	2.90 ± 0.9	Huang et al., 2016 [10]
	21.16 ± 1.7	3.92 ± 3.0	29.18 ± 4.1	2.22 ± 1.0	Abumourad et al., 2013 [59]
*Vibrio* spp.	1.75 ± 0.49	17.77 ± 9.80	3.54 ± 0.53	7.74 ± 3.24	Sapcharoen and Rengpipat (2013) [60]
	3.73 ± 0.60	12.23 ± 3.80	2.22 ± 1.07	16.23 ± 2.33	García et al., 2017 [65]
	2.48 ± 1.01	8.98 ± 2.09	2.48 ± 2.00	18.21 ± 2.22	Huang et al., 2016 [10]
	1.29 ± 0.05	18.08 ± 4.41	3.11 ± 2.12	17.16 ± 5.01	Abumourad et al., 2013 [59]

The data shown are the mean and standard error of the mean, in that order. The indoor and outdoor samples showed a statistically significant difference (*p* < 0.05).

## Data Availability

The data was not utilized for analysis in this manuscript.

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
