# Peer review of "A Review: Microbes and Their Effect on Growth Performance of *Litopenaeus vannamei* (White Leg Shrimps) during Culture in Biofloc Technology System"

_microorganisms, 2024, doi:10.3390/microorganisms12051013_

Round 1

Reviewer 1 Report

Comments and Suggestions for Authors

As this is a review of the literature, one of the major things a reviewer tries to do is assure that the references cited match the conclusion or observation being made. There seems to be a major issue with the references cited vs. the observation of the authors. It is unclear to this reviewer whether this is just a mix up of the references cited or a mistaken understanding of the references being cited.  I strongly suggest the authors go back and make sure the papers cited reflect what is stated in the review.  Some examples follow:

Line 99 cites reference 119 Tremblay et al 2015 to support the idea that bioflocc technology will work week and meet its goals by having species from each category present to properly function. The Tremblay et al. 2015 article discusses 16S rRNA gene tag impacts on sequencing with no discussion of biofloc. 

Line 70 uses references 35 and 87 to support that mammals, fish and humans have all been shown to benefit from effects of gut microbiota on organ development, digestion, and immunological response. Both of these references are about shrimp. More appropriate references should be included that focus on the three animal types discussed in the text.

Line 93 refers to the impact of biofloc on shrimp culture and cites reference 118 which is a study of the impact of live yeast on sea bass.

It is imperative that a review of the literature be tightly referenced and that the references reflect the statements made in the review. The authors need to make sure that there is not some sort of systemic error in the citation manager and revise the paper to assure all the references cited support their statements.

Comments on the Quality of English Language

A review has to read smoothly and convey to the reader the meaning without having to struggle with the word use. The English language usage of this draft is very poor and makes it hard to read.  They authors use singular and plural nouns inappropriately (e.g. line 10 system needs to be plural), and punctuation is misplaced (e.g., line 13 has a period after Probiotics, and probiotics should not be capitalized). Line 18 also has a misplaced period).

Once you have verified all the references and revised the language, it is strongly suggested to get a native speaker to proof your paper for ease of reading.

Author Response

We greatly appreciate your comments and the thorough revision that greatly contributed to the improvement of our manuscript. Please find the detailed responses below. Red color indicated the corresponding revisions in the re-submitted files.

Comments 1: As this is a review of the literature, one of the major things a reviewer tries to do is assure that the references cited match the conclusion or observation being made. There seems to be a major issue with the references cited vs. the observation of the authors. It is unclear to this reviewer whether this is just a mix up of the references cited or a mistaken understanding of the references being cited. I strongly suggest the authors go back and make sure the papers cited reflect what is stated in the review.

Responses 1: We appreciate your comments, and as per your suggestion, I have ensured that all references in the manuscript are corrected in accordance with the instructions provided.

Comments 2: Line 99 cites reference 119 Tremblay et al 2015 to support the idea that biofloc technology will work week and meet its goals by having species from each category present to properly function. The Tremblay et al. 2015 article discusses 16S rRNA gene tag impacts on sequencing with no discussion of biofloc.

Responses 2: Thank you for pointing out this error. This reference was mistakenly added to this place. I have corrected it and have added relative reference.

Comments 3: Line 70 uses references 35 and 87 to support that mammals, fish and humans have all been shown to benefit from effects of gut microbiota on organ development, digestion, and immunological response. Both of these references are for shrimp. More appropriate references should be included that focus on the three animal types discussed in the text.

Responses 3: We are deeply appreciative of your suggestion. This data was out of topic with wrong reference. I have removed this data from my manuscript and added more relevant data with references.

Comments 4: Line 93 refers to the impact of biofloc on shrimp culture and cites reference 118 which is a study of the impact of live yeast on sea bass.

Responses 4: Thank you for bringing this mistake to my attention. The reference included in this line was added erroneously. I have rectified this by removing the incorrect reference and substituting it with the appropriate one, which accurately reflects the source of this information.

Comments 5: It is imperative that a review of the literature be tightly referenced and that the references reflect the statements made in the review. The authors need to make sure that there is not some sort of systemic error in the citation manager and revise the paper to assure all the references cited support their statements.

Responses 5: I have meticulously reviewed all your suggestions and implemented the requisite revisions to accommodate them. Specifically, I have verified that all references in the manuscript are rectified in line with the provided instructions.

Comments 5: A review has to read smoothly and convey to the reader the meaning without having to struggle with the word use. The English language usage of this draft is very poor and makes it hard to read.  They authors use singular and plural nouns inappropriately (e.g. line 10 system needs to be plural), and punctuation is misplaced (e.g., line 13 has a period after Probiotics, and probiotics should not be capitalized). Line 18 also has a misplaced period). Once you have verified all the references and revised the language, it is strongly suggested to get a native speaker to proof your paper for ease of reading.

Responses 5: Thank you sincerely for bringing this issue to my attention. Following your suggestion, I have meticulously proofread my manuscript with the help of a native English speaker. I am confident that this step has significantly enhanced the clarity and readability of the manuscript.

Reviewer 2 Report

Comments and Suggestions for Authors

Include a section with the meta-analysis that you conducted to select the articles referred to in this document.

Abstract

This section is well prepared.

1. Introduction

This section is well prepared.

Lines 71 and 75 “Litopenaeus vanameii” use italics.

Line 77 “800 mg L1” number “1” must be superscript.

2. Microbes in BFT during shrimp culture

This section is well prepared.

Check scientific names which are not in italics.

Include a table with the most important data connecting shrimp production with phyto and zooplankton diversity.

3. White leg Shrimp Microbial Composition

This section is well prepared.

Check scientific names which are not in italics.

4. Association between water and gut microbial communities

This section is well prepared.

Check scientific names which are not in italics.

5. Probiotics

This section is well prepared.

Check scientific names which are not in italics.

6. Microbial Effect on shrimp growth

This section is well prepared; however, I recommend that the authors go deeper into this section and integrate it as “Microbial effect in shrimp physiology, metabolism, and immunology”

Include a brief discussion regarding the advantages and disadvantages of biofloc systems over other culture systems.

7. Conclusion and future perspectives

This section is well prepared.

References

Check the numeration in this section. They must start with the number [1] in the introduction section.

Author Response

We are very grateful for your affirmation of our work. We greatly appreciate your comments and the thorough revision that greatly contributed to the improvement of our manuscript. Please find the detailed responses below. Red color indicated the corresponding revisions in the re-submitted files.

Comments 1: This section is well prepared. Lines 71 and 75 “Litopenaeus vanameii” use italics. Line 77 “800 mg L1” number “1” must be superscript.

Responses 1: Your input is greatly appreciated, and following your recommendation, I have rectified all scientific names by italicizing them for consistency and accuracy.

Comments 2: Microbes in BFT during shrimp culture. This section is well prepared. Check scientific names which are not in italics. Include a table with the most important data connecting shrimp production with phyto and zooplankton diversity.

Responses 2: Thank you for your suggestion. I have made all scientific names as italic, and I have added table related to plankton. (Table.1).

Comments 3: 6. Microbial Effect on shrimp growth. This section is well prepared; however, I recommend that the authors go deeper into this section and integrate it as “Microbial effect in shrimp physiology, metabolism, and immunology.”

Responses 3: Thank you for your suggestions. I have added microbial effect in shrimp related to physiology, metabolism, and immunology from line 673-704.

Comments 4: Check the numeration in this section. They must start with the number [1] in the introduction section.

Responses 4: Your input is greatly appreciated, and following your recommendation, I have commenced all references from [1] in the introduction section as advised.

Round 2

Reviewer 1 Report

Comments and Suggestions for Authors

The review is a vast improvement over the previous version, but still needs to be upgraded to improve its readability and usefulness to the reader. The authors should consider putting in subheadings to clarify specific focuses under each of the major headings. They also need to make sure to make new paragraphs when introducing entirely new ideas or discussion topics - it was very confusing when this was done and, often, with little of no transition.  They should also consider cutting down some of the verbiage and focus it better.  I liked the discussion/conclusion section.

Some specific points:

Line 69 – the convention is to fully spell out the genus species on first use then abbreviate the genus thereafter. So lines 69 and 70 are reversed in this usage. Additionally, line 82 should have the genus abbreviated. Also line 73

Line 72 – convention is to put punctuation after the reference so should read “…investigations [12].” (get rid of extra period)

Line 103 by saying dinoflagellates and diatoms parenthetically you are implying these are the only groups of algae present. Is that what you meant to imply?  I find it hard to believe they are the only two microalgal groups present

Line 113 – Did you mean à “The second-most-dominant phylum was Bacteroidetes at about 21.9%.”  Doesn’t make sense as written.

Line 118 to 124 – I think the Cardona information should be a separate paragraph and have a better transition from the prior paper’s results. Would help if you classed them all at the phylum level for a direct contract to the previous discussion.  You jump from discussing phyla to family to genus and is this really different – help the reader to see the point.

Line 130 – spell out species  “…up of Vibrio species.”  And not one of the dominant phyla (Pseudmonadata) discussed above.

Line 145 – why is reference 85 here? Seems out of sequence but appropriate for what you are discussing.

Line 162 – phytoplankton is a plural noun so should verb should be serve not serves (phytoplankter is singular)

Line 183 – this seems out of place and really adds nothing to the paragraph. Please clarify. Also think Heterokontophyta is a subphylum not the phylum.

Line 190 – be consistent here and list the dinoflagellates as a class as you did with all the other groups.

Line 193-195 – the treatment of ponds with probiotics and results described are not present in the reference 30 that was cited. Did the authors mean reference 33? This needs to be fixed.

Lines 196 to 198 – Again the authors refer to the Nasrullah et al. reference 30 and indicte that a preference by farmers for different algal diversity. However, the reference does not state this and is a screen of different algal populations in East Java cultures. Need to clarify this discussion.

Line 214 – missing comma “..nematodes, rotifers, and..”

Table 1 column 3 – Involved in biofiltration… (and in other sections – change to involved). Don’t capitalize eutrophication (not a proper noun). Be consistent with periods at the end of the sentences (missing a few in this column). Under the Nematode row capitalize the first word in column 3 (Serve)

Line 236 – would change “use” to “consume or metabolize”

Line 253 – I would clarify this statement. At first it sounded like you were saying the ocean only had 3.6 million microbes. But what you really mean is different species of microbe – because there are a whole lot more the 3.6 million microbes in the oceans!  Should have a reference as well.

Line 257 – this section is just tossed in without a transition and is very confusing. The study by Cornejo-Granados (ref 42) is being discussed and the reader has no idea that it has happened. Discuss a difference from the water samples but don’t discuss that this study was a comparison of aquaculture systems with disease pressure and not. Singularly confusing paragraph the needs to be broken up and better integrated to make sense.

Line 302 – fix this sentence.

Line 716-717 – I think there is a missing period after aquaculture.  (“…for aquaculture. Microorganisms…). If not, it does not make sense and fix.

Comments on the Quality of English Language

Some minor punctuation errors and capitalization errors need to be addressed. Otherwise, was much improved.

Author Response

We are very grateful for your affirmation of our work. We greatly appreciate your comments and the thorough revision that greatly contributed to the improvement of our manuscript. Please find the detailed responses below. Red color indicated the corresponding revisions in the re-submitted files.

Comments 1: The review is a vast improvement over the previous version, but still needs to be upgraded to improve its readability and usefulness to the reader. The authors should consider putting in subheadings to clarify specific focuses under each of the major headings. They also need to make sure to make new paragraphs when introducing entirely new ideas or discussion topics - it was very confusing when this was done and, often, with little of no transition.  They should also consider cutting down some of the verbiage and focus it better.  I liked the discussion/conclusion section.

Responses 1: We appreciate your comments, and as per your suggestion, we have addressed all comments by introducing new paragraphs to commence discussions or topics. We believe this will improve readability and usefulness to readers.

Comments 2: Line 69 – the convention is to fully spell out the genus species on first use then abbreviate the genus thereafter. So lines 69 and 70 are reversed in this usage. Additionally, line 82 should have the genus abbreviated. Also line 73. Line 72 – convention is to put punctuation after the reference so should read “…investigations [12].” (get rid of extra period).

Responses 2: Thank you for pointing out this error. We have made all the changes that you suggested. You can see line 69, 70, 73 and 81-82.

Comments 3: Line 103 by saying dinoflagellates and diatoms parenthetically you are implying these are the only groups of algae present. Is that what you meant to imply?  I find it hard to believe they are the only two microalgal groups present. Line 113 – Did you mean à “The second-most-dominant phylum was Bacteroidetes at about 21.9%.”  Doesn’t make sense as written.

Responses 3: We are deeply appreciative of your suggestion. We have rectified all the issues you mentioned in both lines. You can refer to line 103 and 112 to see the corrections.

Comments 4: Line 118 to 124 – I think the Cardona information should be a separate paragraph and have a better transition from the prior paper’s results. Would help if you classed them all at the phylum level for a direct contract to the previous discussion.  You jump from discussing phyla to family to genus and is this really different – help the reader to see the point. Line 130 – spell out species “…up of Vibrio species.” And not one of the dominant phyla (Pseudmonadata) discussed above.

Responses 4: We appreciate your comments, and as per your suggestions, all modifications have been completed. Please review line 116 for the adjustments. Additionally, in line 128-130, we have included the species name.

Comments 5: Line 162 – phytoplankton is a plural noun so should verb should be serve not serves (phytoplankter is singular). Line 183 – this seems out of place and really adds nothing to the paragraph. Please clarify. Also think Heterokontophyta is a subphylum not the phylum. Line 190 – be consistent here and list the dinoflagellates as a class as you did with all the other groups. Line 193-195 – the treatment of ponds with probiotics and results described are not present in the reference 30 that was cited. Did the authors mean reference 33? This needs to be fixed. Lines 196 to 198 – Again the authors refer to the Nasrullah et al. reference 30 and indicte that a preference by farmers for different algal diversity. However, the reference does not state this and is a screen of different algal populations in East Java cultures. Need to clarify this discussion.

Responses 5: Thank you for pointing out this error and we greatly appreciate your suggestion. All these questions pertain to a single paragraph. However, the paragraph was originally irrelevant, so we have substituted it with the most pertinent data. Furthermore, there was a mistaken citation reference, which we have now rectified. Please review the changes made in line 181.

Comments 6: Line 214 – missing comma “..nematodes, rotifers, and..” Table 1 column 3 – Involved in biofiltration… (and in other sections – change to involved). Don’t capitalize eutrophication (not a proper noun). Be consistent with periods at the end of the sentences (missing a few in this column). Under the Nematode row capitalize the first word in column 3 (Serve). Line 236 – would change “use” to “consume or metabolize.”

Responses 6: We value your feedback, and in accordance with your suggestions, we have implemented changes. Please refer to line 204 to see the adjustments made. Additionally, we have capitalized all first words in the table. Furthermore, we have also changed word “use” in line 221.

Comments 7: Line 253 – I would clarify this statement. At first it sounded like you were saying the ocean only had 3.6 million microbes. But what you really mean is different species of microbe – because there are a whole lot more the 3.6 million microbes in the oceans!  Should have a reference as well.

Responses 7: Thank you for your suggestion. The note has been revised and can be found in the revised manuscript, Line 238-239.

Comments 8: Line 257 – this section is just tossed in without a transition and is very confusing. The study by Cornejo-Granados (ref 42) is being discussed and the reader has no idea that it has happened. Discuss a difference from the water samples but don’t discuss that this study was a comparison of aquaculture systems with disease pressure and not. Singularly confusing paragraph the needs to be broken up and better integrated to make sense.

Responses 8: Thank you for your suggestion. We have incorporated the changes based on your recommendations. Please refer to line 242 to see the updates.

Comments 9: Line 302 – fix this sentence. Line 716-717 – I think there is a missing period after aquaculture.  (“…for aquaculture. Microorganisms…). If not, it does not make sense and fix.

Responses 9: Thank you for pointing out these errors. The note has been revised and can be found in the revised manuscript, line 702-703.

Reviewer 2 Report

Comments and Suggestions for Authors

I thank the authors for making all the changes I suggested. I have no further comments.

Author Response

Thank you for taking the time and effort to provide insightful guidance. We are very grateful for your affirmation of our work.